# Data Augmentation by Pairing Samples for Images Classification

## Abstract

Data augmentation is a widely used technique in many machine learning tasks, such as image classification, to virtually enlarge the training dataset size and avoid overfitting. Traditional data augmentation techniques for image classification tasks create new samples from the original training data by, for example, flipping, distorting, adding a small amount of noise to, or cropping a patch from an original image. In this paper, we introduce a simple but surprisingly effective data augmentation technique for image classification tasks. With our technique, named *SamplePairing*, we synthesize a new sample from one image by overlaying another image randomly chosen from the training data (i.e., taking an average of two images for each pixel). By using two images randomly selected from the training set, we can generate $N^2$ new samples from $N$ training samples. This simple data augmentation technique significantly improved classification accuracy for all the tested datasets; for example, the top-1 error rate was reduced from 33.5% to 29.0% for the ILSVRC 2012 dataset with GoogLeNet and from 8.22% to 6.93% in the CIFAR-10 dataset. We also show that our SamplePairing technique largely improved accuracy when the number of samples in the training set was very small. Therefore, our technique is more valuable for tasks with a limited amount of training data, such as medical imaging tasks.

## 1 Introduction

For machine learning tasks, such as image classification, machine translation, and text-to-speech synthesis, the amount of the samples available for training is critically important to achieve high performance by better generalization. Data augmentation, applying a small mutation in the original training data and synthetically creating new samples, is widely used to virtually increase the amount of training data (e.g. Krizhevsky et al. (2012), Szegedy et al. (2015), Fadaee et al. (2017)). There are a wide variety of approaches to synthesizing new samples from original training data; for example, traditional data augmentation techniques include flipping or distorting the input image, adding a small amount of noise, or cropping a patch from a random position. Use of data augmentation is mostly the norm for winning image classification contests.

In this paper, we introduce a simple but surprisingly effective data augmentation technique for image classification tasks. With our technique, named *SamplePairing*, we create a new sample from one image by overlaying another image randomly picked from the training data (i.e., simply taking an average of two images for each pixel). By using two images randomly selected from the training set, we can generate $N^2$ new samples from $N$ training samples. Even if we overlay another image, we use the label for the first image as the correct label for the overlaid (mixed) sample. This simple data augmentation technique gives significant improvements in classification accuracy for all tested datasets: ILSVRC 2012, CIFAR-10, CIFAR-100, and SVHN. For example, the top-1 error rate was reduced from 33.5% to 29.0% for the ILSVRC 2012 dataset with GoogLeNet and from 8.22% to 6.93% in the CIFAR-10 dataset with a simple 6-layer convolutional network. We also conducted data augmentation by overlaying an image picked from outside the training set; this approach also gives some improvements, but our technique, which picks an image to overlay from the training set, gives a far more significant gain. To show that our SamplePairing technique often gives larger benefits when the number of samples in the training data is smaller, we conducted the evaluations by reducing the number of samples used for training with the CIFAR-10. The results showed that our technique yields larger improvements in accuracy when the number of samples are smaller than

the full CIFAR-10 dataset. When we used only 100 samples per label (thus, 1,000 samples in total), the classification error rate was reduced from 43.1% to 31.0% with our technique. Based on these results, we believe that our technique is more valuable for tasks with a limited number of training datasets available, such as medical image classification tasks.

## 2 RELATED WORK

When a model is trained with a small training set, the trained model tends to overly fit to the samples in the training set and results in poor generalization. Data augmentation has been widely used to avoid this overfitting (poor generalization) problem by enlarging the size of the training set; thus, it allows the use of a larger network without significant overfitting. For example, Krizhevsky et al. employ a couple of data augmentation techniques in their Alexnet paper (Krizhevsky et al. (2012)) using random numbers: 1) randomly cropping patches of $224 \times 224$ pixels from the original image of $256 \times 256$ pixels, 2) horizontally flipping the extracted patches, and 3) changing the intensity of RGB channels. Many submissions in the ImageNet Large Scale Visual Recognition Challenge (ILCVRC) employed the first two techniques, which can give a 2048x increase in the dataset size. Our SamplePairing technique is another way to enlarge the dataset size. Since it creates pairs of images, it gives $N^2$ samples from the original $N$ samples. Also, our technique is orthogonal to other data augmentation techniques; as detailed later, our implementation employs the above two basic data augmentation techniques used in Alexnet in each epoch. In this case, we can synthesize $(2048 \times N)^2$ samples in total.

There are many ways to avoid overfitting other than the data augmentation. Dropout (Hinton et al. (2012)) and its variants (e.g. Wan et al. (2013)) are famous techniques used to avoid overfitting by randomly disabling connections during training. Batch normalization (Ioffe & Szegedy (2015)), which intends to solve the problem of internal covariate shift, is also known to be (potentially) effective to prevent overfitting. Our SamplePairing data augmentation is again orthogonal to these techniques and we can employ both techniques to achieve better generalization performance.

Our technique randomly picks an image to overlay from the training set and creates pairs to synthesize a new sample. Our technique is not the first to create a new sample from two randomly picked original samples; existing work such as Chawla et al. (2002) and Wang & Perez (2017) also create a new sample from two samples. SMOTE, Synthetic Minority Over-sampling Technique, by Chawla et al. is a well-known technique for an imbalanced dataset; i.e., the dataset consists of a lot of "normal" samples and a small number of "abnormal" samples. SMOTE does over-sampling of the minority class by synthesizing a new sample from two randomly picked samples in the feature space. Comparing our technique against SMOTE, the basic idea of pairing two samples to synthesize a new sample is common between the two techniques; however, we apply the pairing for the entire dataset instead of for the minority class. Wang and Perez conducted data augmentation by pairing images using an augmentation network, a separate convolutional network, to create a new sample. Compared to their work, we simply average the intensity of each channel of pixels from two images to create a new sample and we do not need to create an augmentation network. Also, our technique yields more significant improvements compared to their reported results.

A paper on a technique called *mixup* (Anonymous (2018)) is submitted to the same conference. Although developed totally independently, mixup also mixes two randomly selected samples to create a new training data as we do in this paper. The differences between our SamplePairing and mixup include 1) mixup overlaying sample labels as well while we use the label from only one sample, 2) they use weighted average (with randomly selected weight) to overlay two samples and 3) they tested tasks other than image classification including speech recognition and Generative Adversarial Network (Goodfellow et al. (2014)). About the label to be used in training, Huszár (2017) pointed out that using the label from only one sample without overlaying the labels from two examples will resulted in the same learning result. This way is simpler than blending two labels and also is suitable for a semi-supervised setting. We also tested overlaying labels from two samples in our SamplePairing and we did not see the differences in the results larger than those due to a random number generator (see appendix).

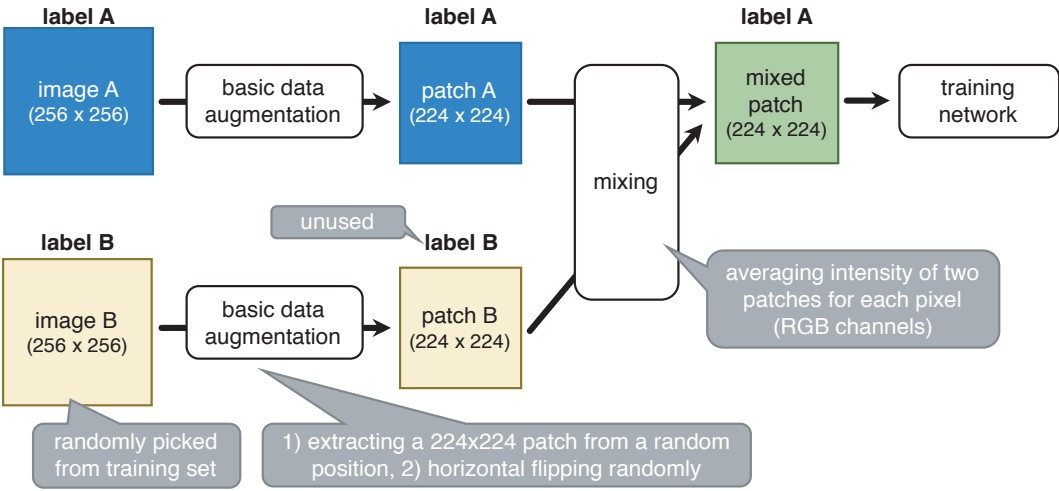

Figure 1: Overview of our SamplePairing data augmentation for ILSVRC dataset. For other datasets, the size of the original images is $32 \times 32$ and the size of the extracted patches is $28 \times 28$.

## 3 METHOD – DATA AUGMENTATION BY *SamplePairing*

This section describes our SamplePairing data augmentation technique. The basic idea of our technique is simple: We synthesize a new image from two images randomly picked from the training set as done in some existing techniques, such as Chawla et al. (2002) and Wang & Perez (2017). To synthesize a new image, we take the most naive approach, just averaging the intensity of two images for each pixel.

Figure 1 depicts the overview of our sample pairing technique. For each training epoch, all samples are fed into the network for training in randomized order. For our technique, we randomly picked another image (*image B* in Figure 1) from the training set, took an average of the two images, and fed the mixed image associated with the label for the first image into the network for training. Hence, essentially, our SamplePairing randomly creates pairs of samples from the training set and synthesizes a new sample from them. The label for the second image was not used in the training. Since two images are equally weighted in the mixed image, a classifier cannot correctly predict the label of the first image (*label A* in Figure 1) from the mixed image unless label A and label B are the same label. So, the training loss cannot become zero even using a huge network and the training accuracy cannot surpass about 50% on average; for $N$-class classification, the maximum training accuracy is $0.5 + 1/(N \times 2)$ if all classes have the same number of samples in the training set. Even though the training accuracy will not be high with SamplePairing, both the training accuracy and validation accuracy quickly improve when we stop the SamplePairing as the final fine-tuning phase of training. After the fine-tuning, the network trained with our SamplePairing can achieve much higher accuracy than the network trained without our technique, as we empirically show in this paper. The network trained with SamplePairing shows higher training error rate and training loss than the network trained without SamplePairing even after the fine tuning since the SamplePairing is a strong regularizer.

When other types of data augmentation are employed in addition to our technique, we can apply them for each image before mixing them into the final image for training. The data augmentation incurs additional time to prepare the input image, but this can be done on the CPU while the GPU is executing the training through back propagation. Therefore, the additional execution time of the CPU does not visibly increase the total training time per image.

As for the entire training process, we do as follows:

1) We start training without our SamplePairing data augmentation (but with basic data augmentations, including random flipping and random extraction).

Table 1: Training and validation sets error rates with and without our SamplePairing data augmentation.

| Dataset | | training error | | validation error | | |
|---|---|---|---|---|---|---|
| | | without SamplePairing | with SamplePairing | without SamplePairing | with SamplePairing | reduction in error rate |
| CIFAR-10 | | 0.55% | 1.25% | 8.22% | 6.93% | -15.68% |
| CIFAR-100 | | 5.78% | 10.56% | 30.5% | 27.9% | -8.58% |
| SVHN | | 0.84% | 2.08% | 4.28% | 4.15% | -3.05% |
| ILSVRC with 100 classes | top-1 error | 0.95% | 3.21% | 26.21% | 21.02% | -19.82% |
| | top-5 error | - | - | 8.58% | 6.11% | -28.74% |
| ILSVRC with 1000 classes | top-1 error | 1.52% | 17.58% | 33.51% | 29.01% | -13.46% |
| | top-5 error | - | - | 13.15% | 11.36% | -13.55% |

2) After completing one epoch (for the ILSVRC) or 100 epochs (for other datasets) without SamplePairing, we enable SamplePairing data augmentation.

3) In this phase, we intermittently disable SamplePairing. For the ILSVRC, we enable SamplePairing for 300,000 images and then disable it for the next 100,000 images. For other datasets, we enable SamplePairing for 8 epochs and disable it for the next 2 epochs.

4) To complete the training, after the training loss and the accuracy become mostly stable during the progress of the training, we disable the SamplePairing as the fine-tuning.

We evaluated various training processes; for example, intermittently disabling SamplePairing with the granularity of a batch or without intermittently disabling it. Although SamplePairing yielded improvements even with the other training processes, we empirically observed that the above process gave the highest accuracy in the trained model.

# 4 EXPERIMENTS

## 4.1 IMPLEMENTATION

In this section, we investigate the effects of SamplePairing data augmentation using various image classification tasks: ILSVRC 2012, CIFAR-10, CIFAR-100, and Street View House Numbers (SVHN) datasets.

For the ILSVRC dataset, we used GoogLeNet (Szegedy et al. (2015)) as the network architecture and trained the network using stochastic gradient descent with momentum as the optimization method with the batch size of 32 samples. For other datasets, we used a network that has six convolutional layers with batch normalization (Ioffe & Szegedy (2015)) followed by two fully connected layers with dropout (Hinton et al. (2012)). We used the same network architecture except for the number of neurons in the output layer. We trained the network using Adam (Kingma & Ba (2014)) as the optimizer with the batch size of 100. During the training, we used data augmentations by extracting a patch from a random position of the input image and using random horizontal flipping as the baseline regardless of whether or not it was with our SamplePairing as shown in Figure 1. In this paper, we do not ensemble predictions in the results. For the validation set, we extracted the patch from the center position and fed it into the classifier without horizontal flipping. We implemented our algorithm using Chainer (Tokui et al. (2015)) as the framework.

## 4.2 RESULTS

We show the improvements in the accuracy by our SamplePairing data augmentation in Table 1. For the ILSVRC dataset, as well as the full dataset with 1,000 classes, we tested a shrinked dataset with only the first 100 classes. For all the datasets we evaluated, SamplePairing reduced the classification error rates for validation sets from 3.1% (for SVHN) up to 28.8% (for the top-5 error rate of the ILSVRC with 100 classes). For training sets, our SamplePairing increased training errors for all datasets by avoiding overfitting. When comparing the ILSVRC with 100 classes and with 1,000

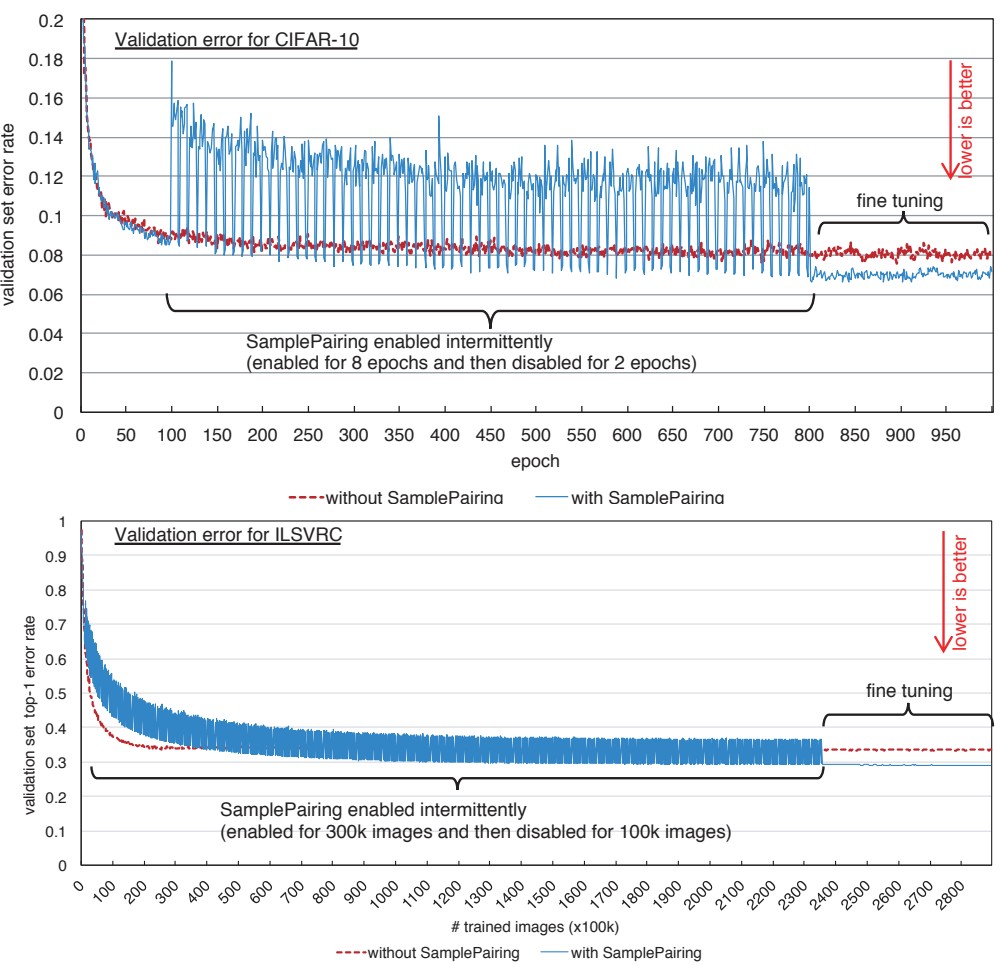

Figure 2: Changes in validation error rates for CIFAR-10 and ILSVRC datasets with and without our SamplePairing data augmentation.

classes (or the CIFAR-10 and CIFAR-100), the case with 100 classes (CIFAR-10) has a much lower training error rate and potentially suffers from the overfitting more severely. Therefore, the benefits of our SamplePairing data augmentation are more significant for the case with 100 classes than the case with 1,000 classes (for the CIFAR-10 than CIFAR-100). These results show that SamplePairing yields better generalization performance and achieves higher accuracy.

Figure 2 and Figure 3 illustrate the validation error rates and the training error rates for the CIFAR-10 and ILSVRC datasets. From Figure 2, we can see much better final accuracy in trade for longer training time. Because we disabled SamplePairing data augmentation intermittently (as described in the previous section), both the training error rates and validation error rates fluctuated significantly. When the network was under training with SamplePairing enabled, both the validation error rate and training error rate were quite poor. In particular, the training error rate was about 50% as we expected; the theoretical best error rate for a 10-class classification task was 45%. Even after the fine tuning, the networks trained with SamplePairing show more than 2x higher training error rate for CIFAR-10 and more than 10x higher for ILSVRC. For the validation set, however, when we disabled the SamplePairing, the validation error rates became much lower than the baseline, which did not employ SamplePairing at all, after the fine tuning as already shown in Table 1.

We also show the changes in training and valiadation losses for ILSVRC dataset in Figure 4. The losses match with the training and validation error rates. Without SamplePairing, we achieved the

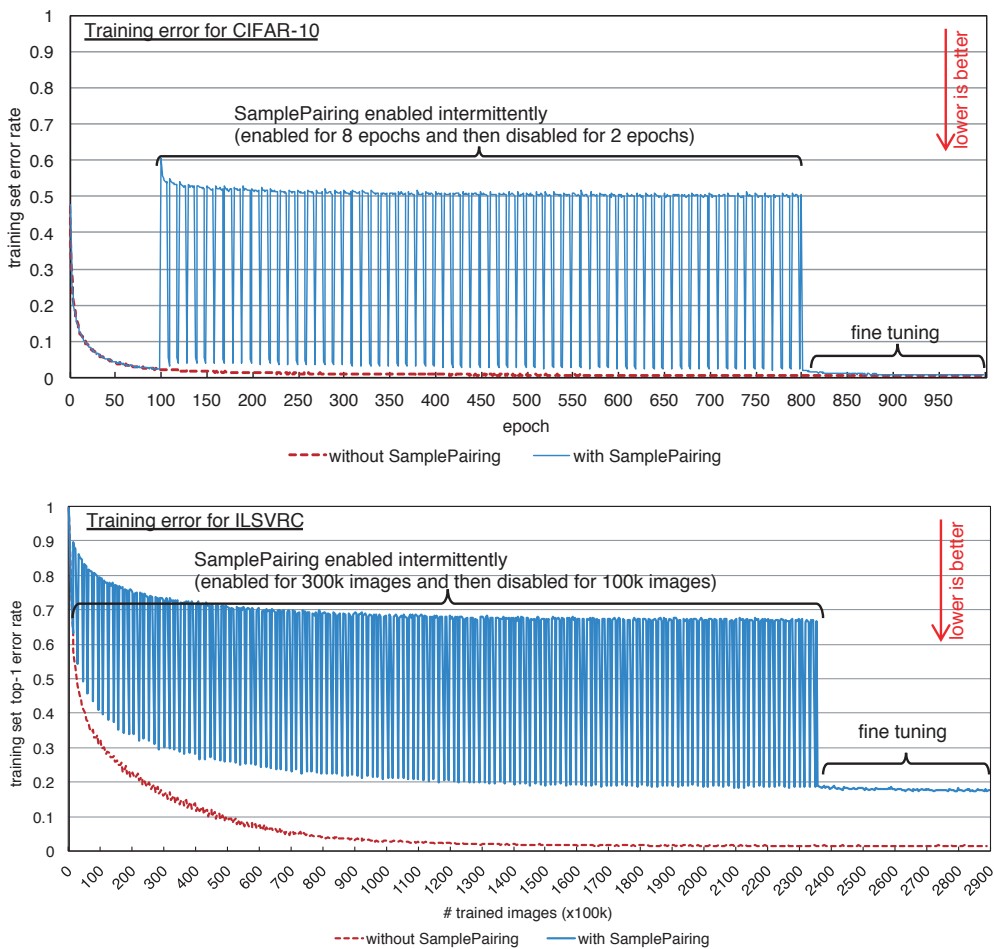

Figure 3: Changes in training error rates for CIFAR-10 and ILSVRC datasets with and without our SamplePairing data augmentation.

minimum validation loss in an early stage of the training and the validation loss gradually increased after that point. With SamplePairing, we did not see such increase in the validation loss.

Figure 5 shows how our SamplePairing data augmentation causes improvements when the number of samples available for training is limited; we change the number of training samples in the CIFAR-10. The CAFAR-10 dataset provides 5,000 samples for each of 10 classes in the training set. We gradually reduced the number of images used for training from 5,000 samples to only 10 samples per class. As shown in Figure 5, the largest gain of SamplePairing was 28% when we used 100 samples per class; the error rate was reduced from 43.1% to 31.0%. When we used only 50 samples per class, the second largest reduction of 27% was achieved. The error rate using 50 samples with SamplePairing was actually better than the error rate using 100 samples without SamplePairing.

When we didn't use all the images in the training set, we also evaluated the effect of overlaying an image randomly chosen from an image not in the training set. For example, if we used only 100 images for training, we made a pool of 100 images selected from the other 4,900 unused images and then randomly picked the image to overlay (i.e., *image B* in Figure 1) from this pool. The results are shown in Figure 5 as "pairing with non-training sample." This method also reduced error rates (except for the case of 10 images per class), but our SamplePairing, which picks an image from the training set, yielded more significant improvements regardless of the training set size. Our SamplePairing, which does not require images other than the training set, is easier to implement and also gives larger gains compared to naively overlaying another image.

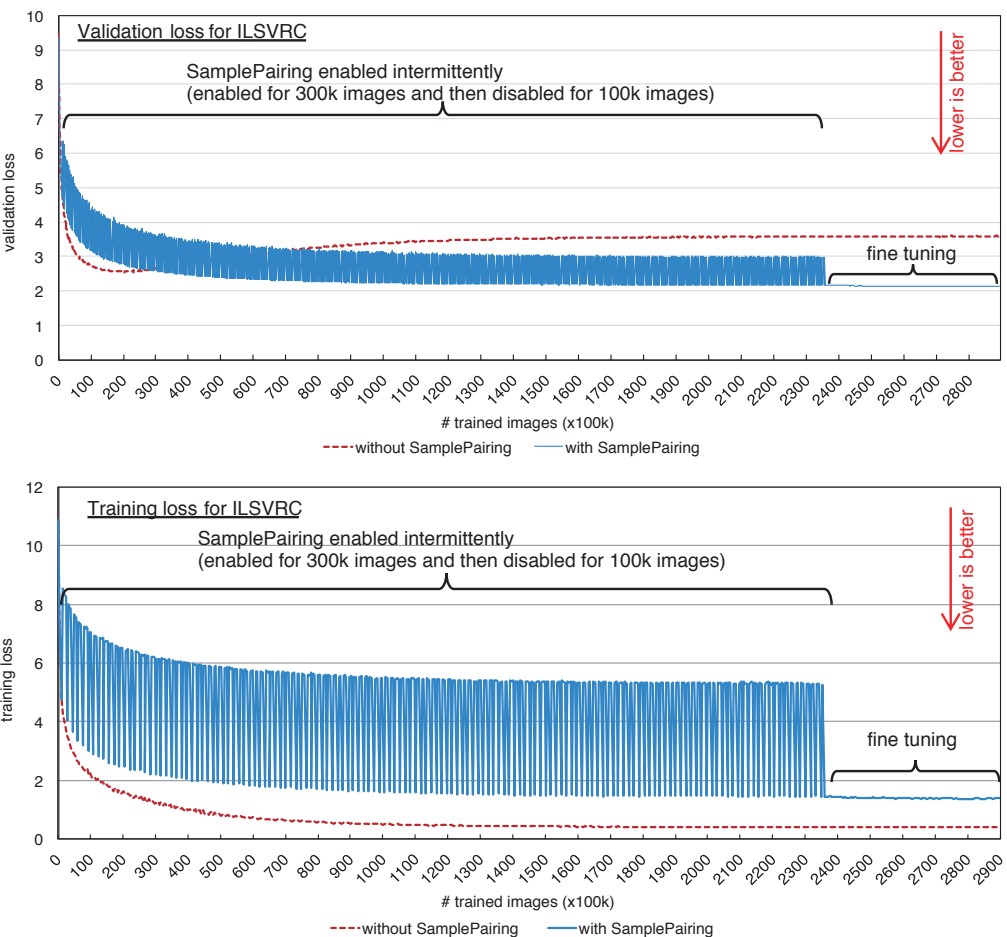

Figure 4: Changes in validation and training losses for ILSVRC datasets with and without our SamplePairing data augmentation.

In this paper, we disabled SamplePairing intermittently. Figure 6 shows how the ratio with and without SamplePairing enabled affects the final classification error on CIFAR-10. In the configuration we used in this paper (disabling SamplePairing in 2 epochs per 10 epochs, i.e. enabled in 80% of the training epochs), we got slightly better final results compared to the case without disabling SamplePairing. However, the differences were not significant if we enabled SamplePairing in more than a half of epochs; hence the tuning of the rate to disable SamplePairing is not that sensitive.

## 5 CONCLUSION

This paper presents our new data augmentation technique named SamplePairing. SamplePairing is quite easy to implement; simply mix two randomly picked images before they are fed into a classifier for training. Surprisingly, this simple technique gives significant improvements in classification accuracy by avoiding overfitting, especially when the number of samples available for training is limited. Therefore, our technique is valuable for tasks with a limited number of samples, such as medical image classification tasks.

In this paper, we only discussed empirical evidences without theoretical explanations or proof. In future work, we like to provide a theoretical background on why SamplePairing helps generalization so much and how we can maximize the benefits of data augmentation. Another important future work is applying SamplePairing to other machine learning tasks especially Generative Adversarial Networks (Goodfellow et al. (2014)).

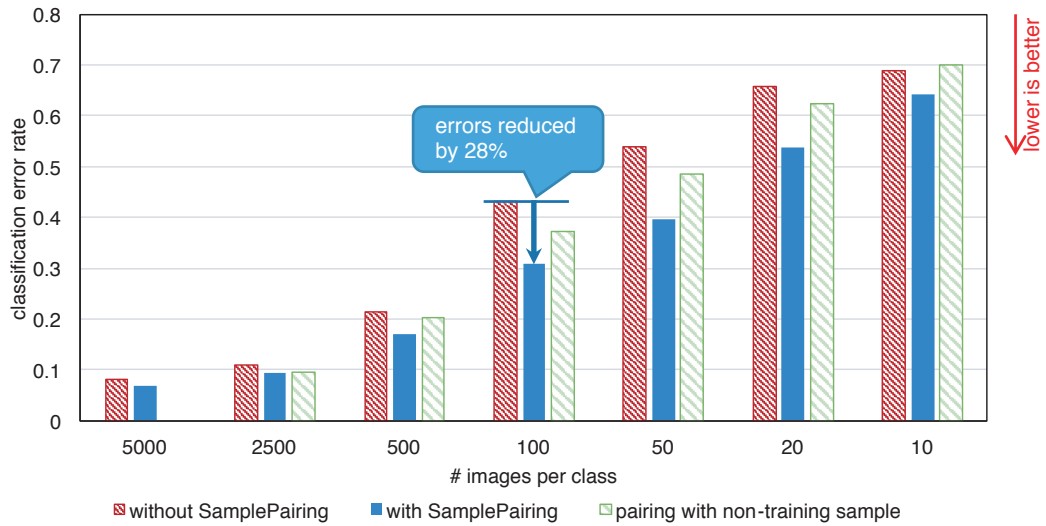

Figure 5: Validation error rates for CIFAR-10 with reduced number of samples in training set.

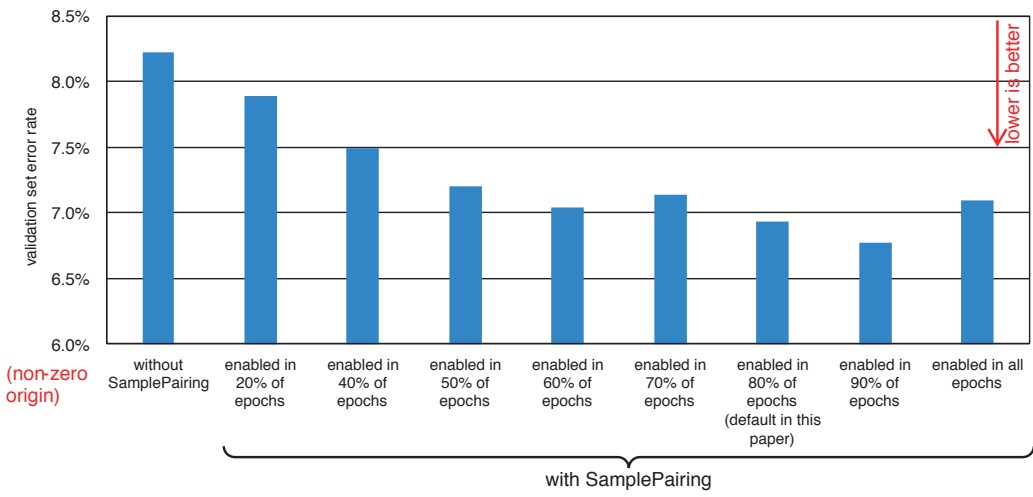

Figure 6: Validation error rates for CIFAR-10 with different ratios for intermittently disabling SamplePairing.

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

## 6 APPENDIX

In our SamplePairing, we take the label from only one sample (*label A* in Figure 1) and discard the label from another (*label B*). Figure 7 shows the validation error with CIFAR-10 if we use both labels for each synthesized sample, i.e. with 0.5 assigned to each in the softmax target of two samples. From Figure 7, we can see that difference due to the use of one label or two labels was not significant. This result is consistent with what Huszár (2017) pointed out on *mixup* (Anonymous (2018)).

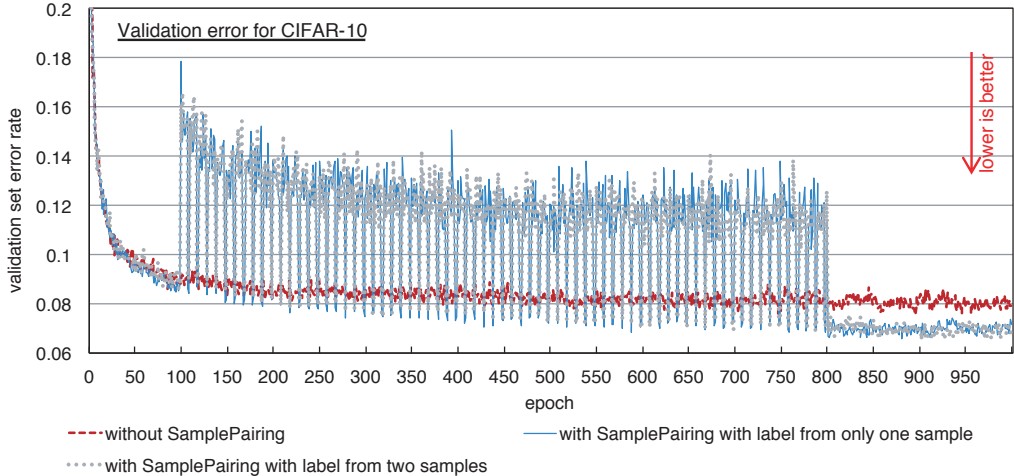

Figure 7: Validation error rates for CIFAR-10 when using lebels from both samples.

For more insight into the improvements by SamplePairing, we show the confusion matrices with and without SamplePairing on CIFAR-10 dataset, which includes four classes of artificial objects (*category A*) and six classes of living thing (*category B*). There are more classification errors within each category (e.g. cat and dog, or automobile and truck) compared to classification errors between two categories (e.g. cat and truck). To detail. Although improvements in error rates varied for targets, SamplePairing reduces both intra- and inter-category classification errors on average; intra-category classification errors were reduced by 10.6% and inter-category errors were reduced by 18.7%.

| Truth | Predicted label | | | | | | | | | |
|---|---|---|---|---|---|---|---|---|---|---|
| | 0) | 1) | 2) | 3) | 4) | 5) | 6) | 7) | 8) | 9) |
| 0) airplane | 908 : 925 | 6 : 5 | 26 : 17 | 12 : 5 | 1 : 3 | 2 : 3 | 1 : 2 | 5 : 3 | 31 : 29 | 8 : 8 |
| 1) automobile | 3 : 1 | 969 : 967 | 1 : 0 | 2 : 2 | 0 : 0 | 0 : 0 | 1 : 0 | 1 : 0 | 5 : 8 | 18 : 22 |
| 2) bird | 13 : 8 | 2 : 0 | 904 : 905 | 22 : 24 | 20 : 20 | 8 : 17 | 18 : 16 | 12 : 3 | 1 : 6 | 0 : 1 |
| 3) cat | 7 : 6 | 2 : 1 | 25 : 18 | 840 : 842 | 23 : 17 | 69 : 85 | 15 : 13 | 12 : 6 | 4 : 9 | 3 : 3 |
| 4) deer | 3 : 2 | 0 : 0 | 25 : 16 | 19 : 15 | 918 : 939 | 8 : 13 | 3 : 8 | 24 : 5 | 0 : 2 | 0 : 0 |
| 5) dog | 3 : 3 | 0 : 0 | 15 : 19 | 71 : 52 | 16 : 9 | 868 : 907 | 4 : 2 | 21 : 5 | 0 : 1 | 2 : 2 |
| 6) frog | 2 : 5 | 0 : 0 | 21 : 8 | 24 : 14 | 8 : 9 | 1 : 5 | 938 : 957 | 2 : 1 | 2 : 0 | 2 : 1 |
| 7) horse | 1 : 3 | 0 : 0 | 8 : 6 | 9 : 14 | 6 : 14 | 10 : 21 | 0 : 0 | 963 : 940 | 0 : 1 | 3 : 1 |
| 8) ship | 16 : 20 | 8 : 7 | 7 : 2 | 5 : 4 | 0 : 0 | 0 : 0 | 1 : 1 | 0 : 0 | 954 : 957 | 9 : 9 |
| 9) truck | 5 : 9 | 38 : 31 | 0 : 1 | 4 : 2 | 1 : 0 | 2 : 0 | 1 : 0 | 0 : 0 | 16 : 6 | 933 : 951 |
| total | 961 : 982 | 1025 : 1011 | 1032 : 992 | 1008 : 974 | 993 : 1011 | 968 : 1051 | 982 : 999 | 1040 : 963 | 1013 : 1019 | 978 : 998 |

† numbers in each cell are (# of samples without SamplePairing :# of samples with SamplePairing)

Figure 8: Confusion matices for CIFAR-10 with and without sample pairing.

