# OpenReview forum: "Data Augmentation by Pairing Samples for Images Classification"
_ICLR.cc/2018/Conference — Reject_

### Official Review · AnonReviewer2 · 2017-11-27
**interesting, but limited contribution**

**Rating:** 4
**Confidence:** 5

**Review:**

The paper proposes a new data augmentation technique based on picking random image pairs and producing
a new average image which is associated with the label of one of the two original samples. The experiments show
that this strategy allows to reduce the risk of overfitting especially in the case of a limited amount of training
samples or in experimental settings with a small number of categories.

+ The paper is easy to read: the method and the experiments are explained clearly.

- the method is presented as a heuristic technique.
1) The training process has some specific steps with the Sample Pairing intermittently disabled.
The number of epochs with enabled or disabled Sample Pairing changes depending on the dataset.
How much si the method robust/sensitive to variations on these choices?
2) There is no specific analysis on the results besides showing the validation and training errors: would it
be possible to see the results per class? Would the confusion matrices reveal something more about the
effect of the method?  Does Sample Pairing help to differentiate similar categories even if they are mixed
at trainign time?
3)  Would it be possible to better control the importance of each sample label rather
than always choosing one of the two as ground truth?

The paper misses an in-depth analysis of the proposed practical strategy.

---

> ### Author Response · Authors · 2018-01-05
> **Author response**
>
> Thank you so much for your comments.
> Please refer the updates in above response on three points you mentioned in the comment.
>
> I like to specially thank the advice on confusion matrix. I have never investigated it.
> On average, SamplePairing gave improvements in classification of similar classes (e.g. two animals or two vehicles) or different classes (e.g. animal and vehicle). But I am doing further investigation on the characteristics of SamplePairing using confusion matrices.

---

### Official Review · AnonReviewer1 · 2017-11-28
**simple technique with nice results, could be analyzed a little deeper**

**Rating:** 5
**Confidence:** 4

**Review:**

The paper investigates a method of data augmentation for image classification, where two images from the training set are averaged together as input, but the label from only one image is used as a target.  Since this scheme is asymmetric and uses quite unrealistic input images, a training scheme is used where the technique is only enabled in the middle of training (not very beginning or end), and in an alternating on-off fashion.  This improves classification performance nicely on a variety of datasets.

This is a simple technique, and the paper is concise and to the point.  However, I would have liked to see a few additional comparisons.

First, this augmentation technique seems to have two components:  One is the mixing of inputs, but another is the effective dropping of labels from one of the two images in the pair.  Which of these are more important, and can they be separated?  What if some of the images' labels are changed at random, for half the images in a minibatch, for example?  This would have the effect of random label changes, but without the input mixing.  Likewise, what if both labels in the pair are used as targets (with 0.5 assigned to each in the softmax target)?  This would mix the images, but keep targets intact.

Second, the bottom of p.3 says that multiple training procedures were evaluated, but I'd be interested to see the results of some of these.  In particular, is it important to alternate enabling and disabling SamplePairing, or does it also work to mix samples with and without it in each minibatch (e.g. 3/4 of the minibatch with pairing augmentation, and 1/4 without it)?

I liked the experiment mixing images from within a restricted training set composed of a subset of the CIFAR images, compared to mixing these images with CIFAR training set images outside the restricted sample (p.5 and Fig 5).  This suggests to me, however, that it's possible the label manipulations may play an important role.  Or, is an explanation why this performs not as well that the network will train these mixing images to random targets (that of the training image in the pair), and never see this example again, whereas by using the training set alone, the mixing image is likely to be repeated with its correct label?  Some more discussion on this would be nice.

Overall, I think this is an interesting technique that appears to achieve nice results.  It could be investigated deeper at some key points.

---

> ### Author Response · Authors · 2018-01-05
> **Author response**
>
> Thank you so much for your comments.
> Please refer the updates 1) and 2) in the above response.
> I am currently implementing SamplePairing in a sub-minibatch granularity. So far, I do not see the significant differences by using smaller granularity of enabling/disabling SamplePairing, e.g. disabling for one mini batch after enabling for four mini batches instead of disabling two epochs after enabling eight epochs. But I am going to add the data with different granularity including the sub-minibatch granularity.

---

### Official Review · AnonReviewer3 · 2017-12-04
**Interesting finding**

**Rating:** 6
**Confidence:** 4

**Review:**

The paper reports that averaging pairs of training images improves image classification generalization in many datasets.
This is quite interesting. The paper is also straightforward to read and clear, which is positive. Overall i think the finding is of sufficient interest for acceptance.

The paper would benefit from adding some speculation on reasons why this phenomenon occurs.
There are a couple of choices that would benefit from more explanation / analysis:  a) averaging, then forcing the classifier to pick one of the two classes present; why not pick both? b) the choice of hard-switching between sample pairing and regular training - it would be interesting if sample-pairing as an augmentation meshed better with other augmentations implementation-wise, so that it could be easier to integrate in other frameworks.

---

> ### Author Response · Authors · 2018-01-05
> **Author response**
>
> Thank you so much for your comments.
> Please refer the updates 1) and 2) in above response on two points you mentioned in the comment (using two labels and switching between SamplePairing and regular training).
> I am adding more experiments on the second point (switching), e.g. using different granularity.  I hope I can add more discussion on this point.

---

### Author Response · Authors · 2017-11-06
**related submission in ICLR 2018**

I found there is another submission discussing a quite similar technique.
mixup: Beyond Empirical Risk Minimization
https://openreview.net/forum?id=r1Ddp1-Rb&noteId=r1Ddp1-Rb

---

### Public Comment · (anonymous) · 2017-11-28
**Clarification**

I am attempting to reproduce the results described in this paper.
I have a few questions:
What is the exact structure of the network trained on the CIFAR-10 dataset?
What fraction of the training data was put aside for the validation set?
Was the training set fabricated by fully using the two basic augmentation techniques (e.g. N samples -> 2048N samples)?
For training on the CIFAR-10 dataset, how many images were used during each SamplePairing epoch and each non-SamplePairing epoch?

Thank you.

---

> ### Author Response · Authors · 2017-11-28
> **Re: Clarification**
>
> Thank you so much for your effort!
>
> > Structure of the network
> (input 28x28x3)
> BatchNorm
> Conv 64
> RELU
> BatchNorm
> Conv 96
> RELU
> MaxPool 2x2
> BatchNorm
> Conv 96
> RELU
> BatchNorm
> Conv 128
> RELU
> MaxPool 2x2
> BatchNorm
> Conv 128
> RELU
> BatchNorm
> Conv 192
> RELU
> MaxPool 2x2
> BatchNorm
> DropOut 40% dropped
> FullConnect 512
> RELU
> DropOut 30% dropped
> FullConnect 10 (100 for CIFAR-100)
> SoftMax
>
> > What fraction of the training data was put aside for the validation set?
> For CIFAR-10, I used 50,000 images included in data_batch_* for training (except for experiments shown in Figure 5). For validation set, I used 10,000 images in test_batch.
>
> > Was the training set fabricated by fully using the two basic augmentation techniques (e.g. N samples -> 2048N samples)?
> Yes. When we test validation images, we extract 28x28 patch from center of the image without ensembling.
>
> > For training on the CIFAR-10 dataset, how many images were used during each SamplePairing epoch and each non-SamplePairing epoch?
> For each epoch (with or without SamplePairing), all 50,000 training images were fed into the training for CIFAR datasets.

---

> > ### Author Response · Authors · 2017-11-28
> > **Re: Clarification**
> >
> > In the above network structure, all convolutions are 3x3 size with padding to keep the size.

---

> > > ### Public Comment · (anonymous) · 2017-11-29
> > > **Re: Re: Clarification**
> > >
> > > Thanks for the quick and detailed reply!
> > >
> > > Just to make sure: were each of 50000 images randomly flipped and cropped (28x28 patch) at a random place before being introduced to the network for each epoch? In other words, are each of the training images slightly altered and therefore different for each non-SamplePairing epoch? Or were each of the 50000 images randomly flipped and cropped before training occured and so that the training set is identical for each non-SamplePairing epoch?

---

> > > > ### Author Response · Authors · 2017-11-29
> > > > **Re: Re: Clarification**
> > > >
> > > > Each image is cropped and random flipped differently for each epoch based on random numbers, not only once before training.

---

> > > > > ### Public Comment · (anonymous) · 2017-12-08
> > > > > **Re: Re: Re: Clarification**
> > > > >
> > > > > Hi, I have a follow up question.
> > > > > Each time you do basic data augmentation, do you generate all possible combinations of patches + flipping, or do you keep the data size to be the same as N (original sample size)?

---

> > > > > > ### Author Response · Authors · 2017-12-09
> > > > > > **Re: Re: Re: Clarification**
> > > > > >
> > > > > > In each epoch, we generate one (but not all) sample for each input sample. Since we use random number generator, the generated patches are different for epoch by epoch. The size of the extracted patch (i.e. input of the classifier) is 28x28 for CIFAR, not the original image size of 32x32, as you can see in above network design.

---

### Public Comment · (anonymous) · 2017-12-06
**Reproducibility**

Hello,

My team and I are attemption to reproduce the results of your paper and had a few queries:

1) Is any code available for the experiments you performed?
2) For the baseline results (without sample pairing) on the CIFAR-10 and CIFAR-100, did you use any augmentation methods such as flipping and cropping the images or simple feed in the raw images?
3) What loss function did you use for the CNN training?

---

> ### Author Response · Authors · 2017-12-06
> **Re: Reproducibility**
>
> Thank you so much for your effort for reproducing our results.
> 1) I am sorry, but not yet published.
> 2) Yes, the baseline uses the flipping and cropping. I will make the paper more clearer on this point.
> 3) I use softmax_cross_entropy function provided by Chainer framework. (http://docs.chainer.org/en/stable/reference/generated/chainer.functions.softmax_cross_entropy.html)

---

> > ### Public Comment · (anonymous) · 2017-12-12
> > **The fine tuning part**
> >
> > Thank you for your reply. I had a further query about the fine tuning part. Can you describe what steps you took during that phase? Did you just let the model train on the original cropped data for that duration (because we don't see any spikes representative of the sample paired data)?

---

> > > ### Author Response · Authors · 2017-12-12
> > > **Re: The fine tuning part**
> > >
> > > In fine tuning part, I just stop applying SamplePairing. The basic data augmentations, drop out etc are still active during the fine tuning phase.

---

> > > > ### Public Comment · (anonymous) · 2017-12-13
> > > > **Augmentation per epoch**
> > > >
> > > > Perfect, than you so much for your responses. One last part we want to get right is whether the augmentations change per epoch for the baseline? As in do you reflip and recrop to create new data for every epoch or just do it once and keep training on that data?

---

> > > > > ### Author Response · Authors · 2017-12-13
> > > > > **Re: Augmentation per epoch**
> > > > >
> > > > > All augmentations (crop, flip, pairing) are per epoch based on random numbers.

---

### Public Comment · (anonymous) · 2017-12-15
**Reproducibility Report for Data Augmentation by Pairing Samples for Images Classification**

Hello everyone,

My team and I aimed to reproduce the results as presented in the paper, under the ICLR 2018 Reproducibility Challenge. Due to limited computation resources, we have only reproduced the paper for the CIFAR-10 dataset.

We reproduced almost similar trends as produced by the paper for training and validation dataset with 5000 samples per class(Refer Report).The validation error reduced significantly in the fine-tuning phase as claimed by the paper. The training error in case of SamplePairing came out to be comparatively higher than that without SamplePairing, hence proving that SamplePairing avoids overfitting. In the paper, it is mentioned that the validation error for CIFAR-10 datasets is decreased by 15.68 %. As the paper hasn't mentioned anything about samples per class in the table for CIFAR-10 we assume them to be taking full dataset with 5000 samples per class. When we reproduced the procedure, we got a reduction in validation error rate by 16.61% which is pretty similar when compared with the result given in the paper. However, for the dataset with smaller samples per class, this particular graph became pretty irregular as we proceeded due to limited dataset and overfitting (See APPENDIX in the Report). Although the final trend in all the samples per class is decreased validation error when trained with SamplePairing, there is an exception in one dataset where we took 500 samples per class. That variance might have arrived because of changes in batch size for lower samples per class. The paper hasn’t mentioned explicitly about the batch size for smaller samples per class, which made us experiment with different values. Although we were able to produce the similar trends for 5000, 2500, 100, 20, and 10 samples per class respectively. We are not able to produce a similar trend for 500 samples per class even after experimenting with a number of different batch size. The relation of trends between SamplePairing within the test set and outside the test set keep on varying with for different batch size and hence no conclusion can be drawn from that. The decrease in validation error for 100 samples per class in our implementation is in accordance with the trend mentioned in the paper, however, the value is not too similar. In the paper,  a 28% reduction in validation error rate is there however, we got a fairly low error reduction i.e. 10.08%. It was said that SamplePairing within the training data produced more effective results. This applied to our results too but not in all cases, as the validation error of SamplePairing outside the training dataset is often higher but in some cases almost similar or lower to validation error of SamplePairing within the training dataset.

We have made a detailed analysis and put it all together in a report. Please access it here https://goo.gl/kN27Cp

---

### Public Comment · (anonymous) · 2017-12-16
**Reproducibility Report**

Many machine learning algorithms are limited by the availability and the amount of data. Running a deep neural network on a small dataset generally results in overfitting without careful adaptation, and does not generalize well for unseen data. This translates to the loss of the algorithm's predictive power. Data augmentation techniques are common to circumvent this problem. Basic data augmentation such as adding noise, randomly cropping and flipping patches of pixels have shown to reduce overfitting and increase the overall model's robustness to noise.

This paper introduces a novel data augmentation technique called SamplePairing for image classification tasks. For each image during training, another image is randomly sampled with replacement from the same training set, and the pairs of images are merged as input. The label of the original image is set as the ground truth label for the newly created sample. This data augmentation technique is simple, and it does not require additional data outside of the original one. The latter is absolutely crucial for small datasets.

Our team mainly focused on reproducing the relevant results for the CIFAR-10 dataset. This is a well studied dataset and is readily available online with detail description as well as instruction to extract it. The source code for SamplePairing technique is not available in the paper, but can be easily implemented. The architecture of the 6 layers convoluted neural networks (CNN) used is also not presented in the paper, but is released by the authors after inquiring on open review . All of the algorithms are implemented in Python. Keras was used for implementing the CNN instead of the Chainer framework used by the authors; but, since both API's are extremely similar, we do not think this decision would have an impact on our results. A major challenge for reproducing the experiment is that all of the data augmentation are determined through random numbers. And since no details about the random number generators (random seed) are communicated by the authors, to obtain the exact same results would require us to perform all of the possible combinations of data augmentation on each sample, which is unfeasible. Another challenge is the computational resources and time that these experiments required. To train on the full dataset, roughly 24 hours was needed with a Nvidia Tesla P100 GPU. The details of how the authors validate their results were also missing, as the nature of the validation error rates presented in the table is unknown. A rather uncommon practice for validating the performance was also employed by the authors, where the entire training set is used as input for the CNN, and the entire testing set is used as validation set.

After carefully following the procedures presented in the paper, we obtained a final validation accuracy at the last epoch of 90.89% for the full dataset with and without using SamplePairing. A slight improvement can be observed, however, if we investigate the past 20 epochs, which yielded a 0.41% increase of accuracy on average. One significant difference in the behavior of validation error rates during each epochs in SamplePairing phase is that the gradual decrease in error rates was not present in our results, instead the validation error rates kept relatively constant. Our validation error rates were also much higher than that presented in the paper throughout the SamplePairing phase. The effect of applying SamplePairing was also studied for datasets that have smaller numbers of samples per class. The subsets are extracted from the original dataset with 2500, 500, 100, 20, and 10 samples per class, randomly and respectively. Based on the accuracy, only the dataset with 20 samples per class yielded a better accuracy when SamplePairing is used, the other subsets have a poorer performance when SamplePairing is applied.

Overall, the paper presents the SamplePairing technique, and the training procedures in a clear and concise manner. It is easy to read even for readers that are not familiar with the relevant literature. The results of the paper can be interpreted straight forwardly with figures, where the effects of applying SamplePairing is strongly contrasted. Based on the results, by using this data augmentation technique, it can help reduce overfitting and even obtain reasonable results for small datasets.

The lack of source code for this paper greatly contributed to the difficulty of reproducing the same results. Despite of the fact that our results do not all agree with what is presented in the paper, we believe that with more fine tuning of CNN's hyperparameters, and more experiments, we can achieve the same conclusion as what is presented in the paper.

---

### Public Comment · (anonymous) · 2017-12-16
**Reproducibility challenge report**

We attempted to reproduce the results of this paper. The authors were quite clear in the implementation procedure for the Samplepairing technique as a data augmentation method. They clearly described the steps they took for their experiments. Due to unavailability of published code, created our own Samplepairing, cropping and flipping methods. We choose the CIFAR-10 dataset for our reproducibility challenge given its relatively low computational complexity.

Our results coincide with the author’s in that Samplepairing does improve the classification performance of the classifier on the validation set. However, we were unable to achieve the levels of accuracy stated in the paper with the CNN architecture. Our validation error rate for the classifier trained on Samplepaired data was 0.139, a 33.8% improvement over the baseline result. Naturally, this different could be down to assumptions we made about unknown factors in the experimental procedure particular pertaining the 6 layered CNN architecture as well as the random nature of the augmentation techniques.  However, we were able to achieve similar results on the validation set using the same training horizon and classifier on the original 32 by 32 CIFAR-10 dataset without any data augmentations. It would certainly have been helpful if a comparison without any augmentation techniques would have been used as a control.  More details about the CNN architecture used on the smaller datasets as well as what steps the authors took to finetune the classifier would also have made the results easier to reproduce. Computational cost was also a factor considering the large training horizons and the large number of augmentations per epoch, mkaing reproducing the results all the more challenging.

Despite the difference, our results also show that Samplepairing helps improve validation performance, lowering variance at the cost of higher bias as displayed by the higher training error. We believe that with finetuning our model would yield accuracies close to what the author's saw in their experiments.  The paper itself was concise and very explicit about the details pertaining to the Samplepairing methodology as well as the augmentation techniques used which certainly helped in reproducing it.

The detailed report of our analysis is accessible on :https://drive.google.com/open?id=1bVwqbcQXVkNRju2Schi_oPHS5p_VAAJp

---

### Author Response · Authors · 2018-01-05
**Author response**

First of all, we greatly thank the reviewers for their valuable comments. Also, we like to thank who made effort to reproduce our results.
I updated the submission based on the comments from reviewers.

The major updates are:
1) I added discussion on mixup, which is proposed in another submission (https://openreview.net/forum?id=r1Ddp1-Rb&;noteId=r1Ddp1-Rb), in related work.
Although mixup does blending two samples as we do in this paper, mixup also blends labels from both samples while we pick only one.
There is a blog post by Ferenc Huszár (http://www.inference.vc/mixup-data-dependent-data-augmentation/), which points out that using label from one sample will give the same results by reformulating the loss function of mixup.
I also tested using both labels in our SamplePairing and it did not give significant difference as show in Figure 7 (in Appendix).

2) In this paper, we intermittently disable SamplePairing in 20% of the epochs. I added Figure 6 on how this ratio affects the final results to answer the reviewers' questions. By intermittently disabling SamplePairing, we can get small improvements compared to the case without disabling SamplePairing. But this improvement is minor compared to the improvements by SamplePairing itself; hence the training with SamplePairing is not so sensitive to this (potentially workload dependent) tuning parameter.

3) I added confusion matrices with and without SamplePairing to show how samples in each class are predicted in Figure 8 (in Appendix).

---

### Decision · Program_Chairs · 2018-01-29
**ICLR 2018 Conference Acceptance Decision**

**Decision:**

Reject

**Comment:**

The paper proposes a data augmentation technique for image classification which consists in averaging two input images and using the label of one of them. The method is shown to outperform the baseline on the image classification task, the but evaluation doesn’t extend beyond that (to other tasks or alternative augmentation mechanisms); theoretical justification is also lacking.